

# miR-138-5p ameliorates intestinal barrier disruption caused by acute superior mesenteric vein thrombosis injury by inhibiting the NLRP3/HMGB1 axis

Yuejin Li[*], Ping Ling[*], Yu Li, Yongzhi Wang, Guosan Li, Changtao Qiu, Jianghui Wang and Kunmei Gong

The First People's Hospital of Yunnan Province, The Affiliated Hospital of Kunming University of Science and Technology, Kunming, China

[*] These authors contributed equally to this work.

## ABSTRACT

**Background.** Acute superior mesenteric venous thrombosis (ASMVT) decreases junction-associated protein expression and intestinal epithelial cell numbers, leading to intestinal epithelial barrier disruption. Pyroptosis has also recently been found to be one of the important causes of mucosal barrier defects. However, the role and mechanism of pyroptosis in ASMVT are not fully understood.

**Methods.** Differentially expressed microRNAs (miRNAs) in the intestinal tissues of ASMVT mice were detected by transcriptome sequencing (RNA-Seq). Gene expression levels were determined by RNA extraction and reverse transcription-quantitative PCR (RT–qPCR). Western blot and immunofluorescence staining analysis were used to analyze protein expression. H&E staining was used to observe the intestinal tissue structure. Cell Counting Kit-8 (CCK-8) and fluorescein isothiocyanate/propidine iodide (FITC/PI) were used to detect cell viability and apoptosis, respectively. Dual-luciferase reporter assays prove that miR-138-5p targets NLRP3.

**Results.** miR-138-5p expression was downregulated in ASMVT-induced intestinal tissues. Inhibition of miR-138-5p promoted NLRP3-related pyroptosis and destroyed tight junctions between IEC-6 cells, ameliorating ASMVT injury. miR-138-5p targeted to downregulate NLRP3. Knockdown of NLRP3 reversed the inhibition of proliferation, apoptosis, and pyroptosis and the decrease in tight junction proteins caused by suppression of miR-138-5p; however, this effect was later inhibited by overexpressing HMGB1. miR-138-5p inhibited pyroptosis, promoted intestinal epithelial tight junctions and alleviated ASMVT injury-induced intestinal barrier disruption *via* the NLRP3/HMGB1 axis.

# INTRODUCTION

Mesenteric venous thrombosis (MVT) is an infrequent thrombotic event that can cause devastating intestinal hemorrhagic ischemia, peritonitis, sepsis and shock (*Acosta et al., 2008*; *Blumberg & Maldonado, 2016*). Acute MVT (AMVT) represents 6% to 9% of all

Corresponding author
Kunmei Gong, kunhuagongkunmei@163.com

mesenteric ischemia cases (*Singal, Kamath & Tefferi, 2013*). Superior AMVT (ASMVT) is the most common type of AMVT (*Brunaud et al., 2001*). Despite advances in managing thromboembolic diseases over the past 40 years, the rates of readmission and average 30-d mortality of ASMVT are still high in severe cases (*Andraska et al., 2020*). Therefore, the discovery of a novel and effective treatment strategy for ASMVT is vital.

The intestinal epithelial barrier is an important barrier for the body to resist the invasion of pathogens and microorganisms and is associated with a number of disease states, both intestinal and systemic (*Odenwald & Turner, 2017*). A previous study reported that ASMVT decreased junction-related protein expression and intestinal epithelial cell number and increased permeability, bacterial translocation and inflammatory cytokines over a time course (*Dokladny, Zuhl & Moseley, 2016*; *Xu et al., 2020*; *Yang et al., 2018*). These risk factors are strongly associated with the pathogenesis of intestinal epithelial barrier disruption (*Dokladny, Zuhl & Moseley, 2016*). Pyroptosis, an inflammatory programmed cell death, is a cause of mucosal barrier defects, including intestinal epithelial barrier defects (*Jia et al., 2020*; *Xu et al., 2020*). However, the role of pyroptosis in ASMVT is not completely understood. It is important to investigate the effects and potential mechanisms of pyroptosis on intestinal epithelial barrier disruption in ASMVT.

The inflammasome is a cytosolic monitoring protein complex that can lead to caspase-1 activation, proinflammatory cytokine maturation and cell pyroptosis (*Ahn et al., 2018*). The nucleotide-binding domain leucine-rich repeat family protein 3 (NLRP3) inflammasome is present primarily in immune and inflammatory cells following activation by inflammatory stimuli, such as interleukin-1$\beta$ (IL-1$\beta$) and IL-18 (*Martinon, Burns & Tschopp, 2002*). Studies have reported that NLRP3 inflammasome activation mediates impaired tight junction protein expression, which disrupts intercellular tight junctions (*Ma et al., 2020*; *Zhuang et al., 2015*). These inflammatory factors and tight junction proteins are strongly associated with the intestinal epithelial barrier (*Kaminsky, Al-Sadi & Ma, 2021*; *Schuhmann et al., 2011*). However, the activity of the NLRP3 inflammasome is unclear in ASMVT. It has been reported that pyroptosis is regulated by the NLRP3 inflammasome (*Zheng & Kanneganti, 2020*), and NLRP3-related pyroptosis plays an important role in intestinal diseases (*Chen et al., 2019*; *Huang et al., 2019*). In addition, research has reported that the downregulation of junction proteins in endothelial cells is attributed to high-mobility group box protein 1 (HMGB1) release (*Chen et al., 2015b*; *Lian et al., 2019*). Thus, we speculated that the NLRP3 inflammasome and HMGB1 release promote intestinal epithelial barrier disruption by increasing intestinal epithelial cell pyroptosis and decreasing tight junctions.

microRNAs (miRNAs) are a class of small noncoding RNAs that function in the posttranscriptional regulation of gene expression (*Saliminejad et al., 2019*). A correlation between miRNA expression is associated with autophagy, inflammation, oxidative stress, apoptosis, and epithelial barrier function (*Akbari, 2020*). For example, miR-381-3p, miR-146a and miR-34a-5p regulate intestinal epithelial cell proliferation and barrier function after intestinal ischemia/reperfusion injury by targeting nuclear receptor-related protein 1, toll-like receptor 4 and sirtuin 1, respectively (*He et al., 2018*; *Liu et al., 2018*; *Wang et al., 2016*). However, whether the changes in miRNA expression after ASMVT and whether it

affects intestinal epithelial barrier function by regulating the NLRP3 inflammasome need to be further investigated.

In our preliminary study, we observed that miR-138-5p was expressed at low levels in ASMVT-induced intestinal tissues by RNA-Seq (RNA Sequencing) analysis (the statistical efficacy of this experimental design, calculated as RNA SeqPower, was 0.56 for each group of three samples, with at least three replicates of each sample). Overexpression of miR-138-5p inhibited NLRP3-related pyroptosis and promoted tight junctions between intestinal epithelial cells. However, overexpression of miR-138-5p ameliorated ASMVT injury-induced intestinal barrier disruption in rats.

## MATERIALS & METHODS

### Animal model and treatment

Six- to eight-week-old male Sprague Dawley rats (weight, 300–350 g) were supplied by Hunan SJA Laboratory Animal Co., Ltd. and housed individually in a temperature-controlled barrier facility with light-dark (12:12) cycles at 22 °C with 50% humidity and received ad libitum food and water. The animal experiments were approved by the Animal Ethics Committee of Kunming University of Science and Technology.

The ASMVT rat model was established on the foundation of a previous study (*Badripour et al., 2022*). The rats were fasted for 12 h before the operation and were given free access to water. In the lower abdomen of the rat, the rat was anesthetized using intraperitoneal injection of ketamine (90 mg/kg) in combination with xylazine (12 mg/kg), and a midline incision of approximately two cm in length was made in the upper abdomen. The position of the cecum was determined after entering the abdominal cavity. The small intestine was pulled five cm away from the ileocecal region, and the mesentery was exposed. The main branches of the mesenteric vein and the ends of the arch vein were ligated with 7-0 nylon thread. During mesenteric vein clamping, approximately 15–20 ml/kg saline was injected intraperitoneally intermittently to prevent transient hypovolemia after loosening of the arterial clamp. Mesenteric vein dilatation and intestinal congestion were dark red, indicating successful modeling. After closing the abdomen, 25–30 m/kg saline (containing gentamicin 320,000 U/L) was added into the abdominal cavity to prevent abdominal surgical infection. After surgery, the rats were placed back into the feeding cage for observation, one rat per cage, and were allowed to eat and drink freely.

Thirty rats were randomly divided into the following groups: (1) Sham group: after surgical exposure of the superior mesenteric vein; (2) ASMVT group; (3) ASMVT+miR-138-5p mock group: the rats received 100 nM miR-138-5p mimic treatment (intraperitoneal injection, 0.5 mL). Rats were sacrificed by the spinal dislocation method under deep anesthesia. The affected intestine was cut down from the middle, and the intestinal contents were washed with normal saline. Small intestinal tissues were stored at −80 °C until use.

### Animal care, feeding, housing, and enrichment

The rat feeding room was disinfected twice a week with 0.1% peroxyacetic acid spray, and the rat box and drinking water bottles were soaked in 0.2% peroxyacetic acid for 3 min or

autoclaved every month. Rat feed was added according to the principle of small amount and multiple times, soft food was changed daily, and drinking water was autoclaved water and melon seeds were fed to ensure the welfare of experimental animals during the experiment. One week before the surgery, we started one cage for one rat and observed the rats' food and water intake, activity level, whether the eyes were alert, tail color, etc., and recorded the temperature, humidity and ventilation of the feeding room. After the surgery, the rats were kept in one cage and allowed to eat and drink freely, and the condition of the rats was observed every day.

In this study, the experiment was divided into three groups, and considering the failure of the experiment in the modeling process and the death of the rats, 10 rats were needed for each group, and the total number of experimental rats needed was 30. The number of samples was calculated based on (the method of comparison between two groups was chosen): the formula $N = 2[(a+b)^2 \times \sigma^2]/(\mu1-\mu2)^2$ was chosen to calculate the number of rats.

## Details of euthanasia method(s) used

When loading the rat into the container, the animal is induced with a low concentration of carbon dioxide and then switched to a high concentration to cause rapid unconsciousness. When the rat appears to be dead, the gas infusion is continued for 2 min, and the rat must be confirmed dead before it is removed from the euthanasia container. After the rat is removed, the rat is confirmed dead by methods such as decannulation.

## Criteria established for euthanizing animals prior to the planned end of the experiment and whether this was needed

The rats were unable to eat, were depressed and trembling after surgery, showed anxiety, decreased physical condition, ulceration of body surfaces, and extreme behavior during the experiment and were euthanized using carbon dioxide to reduce nonessential pain.

## What happened to any surviving animals at the conclusion of the experiment

The experimental endpoint of the animals was to remove the small intestine of the rats to observe the pathological condition and to make sections for subsequent experiments. Therefore, our rats were sacrificed before the small intestine was removed, and no rats survived at the end of the experiment.

## Hematoxylin and eosin (H&E) staining

Tissues were harvested and subjected to 4% paraformaldehyde (PFA) fixation for 24 h. The tissue blocks were sectioned at 5-$\mu$m thickness for subsequent staining. These sections were stained with hematoxylin-eosin (H&E) for 10 min at room temperature. H&E was observed by a Nikon Eclipse 80i microscope (Nikon Corporation).

## Immunofluorescence

These sections and cells were fixed with 4% paraformaldehyde for 15 min and blocked with 3% BSA in phosphate buffer solution (PBS) for 30 min at room temperature. Then, the cells were incubated with rabbit antibodies against NLRP3 (dilution 1:500; cat. no.,

ab263899; Abcam, Cambridge, UK), HMGB1 (dilution 1:500; cat. no., ab18256; Abcam, Cambridge, UK), ZO-1 (5 μg/ml; cat. no., 40-2300; Thermo Fisher Scientific, Waltham, MA, USA), occludin (dilution 1:100; cat. no., ab216327; Abcam, Cambridge, UK) and Claudin-18 (dilution 1:500; cat. no., ab203563; Abcam, Cambridge, UK) overnight at 4 °C and Alexa Fluor® 488 goat anti-rabbit IgG secondary antibody (dilution 1:400; cat. no., dilution 1:500; cat. no., ab203563; Abcam, Cambridge, UK) for 1 h at room temperature. The cell nucleus was stained using 0.1% DAPI for 5 min at room temperature. Staining was observed by a Nikon Eclipse 80i microscope (Nikon Corporation, Tokyo, Japan).

## Western blot

The tissues and cells were isolated and denatured *via* RIPA and BCA Protein Determination (Pierce Biotechnology, Waltham, MA, USA) for total protein extraction and quantification, respectively. Protein samples were separated by 10% SDS–PAGE and transferred to polyvinylidene difluoride membranes. After blocking in 5% skim milk. Subsequently, the membrane was incubated with primary antibodies: NLRP3 (1:1000; cat. no. ab26389; Abcam, Cambridge, UK), Caspase-1 (1:1000; cat. no. ab286125; Abcam, Cambridge, UK), ASC (1:500; cat. no. ab180799; Abcam), GSDMD (1:1000; cat. no. ab219800; Abcam, Cambridge, UK), IL-18 (1:1000; cat. no. ab191860; Abcam, Cambridge, UK), IL-1$\beta$ (1:1000; cat. no. ab254360; Abcam, Cambridge, UK), HMGB1 (dilution 1:1000; cat. no., ab18256; Abcam), ZO-1 (1 μg/ml; cat. no., 40-2300; ThermoFisher Scientific, Waltham, MA, USA), occludin (dilution 1:1000; cat. no., ab216327; Abcam, Cambridge, UK) and Claudin-18 (dilution 1:1000; cat. no., ab203563; Abcam, Cambridge, UK) and GAPDH (1:5000; cat. no. ab8245; Abcam, Cambridge, UK) antibodies were applied to membranes, followed by HRP-conjugated secondary antibody (1:5000; cat. no. ab20272; Abcam, Cambridge, UK). Antibody binding was detected by enhanced chemiluminescence reagent (Thermo Fisher Scientific, Waltham, MA, USA). ImageJ (V version 1.47; National Institutes of Health) was used to analyze the gray value of each band on the membrane.

## miRNA sequencing

Total RNA was extracted from three tissues from each group. The RNA amount and purity of each sample were quantified using a NanoDrop ND-1000 (NanoDrop, Wilmington, DE, USA). The RNA integrity was assessed by Agilent 2100 with RIN >7.0. The small RNA sequencing library was prepared by TruSeq Small RNA Sample Prep Kits (Illumina, San Diego, CA, USA). Next, deep sequencing was performed with an Illumina HiSeq2500 following the vendor's recommended protocol. The differentially expressed miRNAs were selected with statistical significance ($p$ value <0.05) by using Student's t test. Two computational target prediction algorithms (TargetScan, v5.0, and Miranda, v3.3a) were used to determine which genes are targeted by the most abundant miRNAs. We then calculated overlaps between the predicted data from both algorithms. Additionally, these most abundant miRNAs and miRNA targets were annotated with GO terms and KEGG pathways. An RNA-Seq analysis was conducted by Hangzhou Lianchuan Biotechnology Co., Ltd. (Hangzhou, China).

## RNA extraction and reverse transcription-quantitative PCR (RT–qPCR)

TRIzol® reagent (Invitrogen; Thermo Fisher Scientific) was used to extract total RNA from tissues and cells. In this study, the cDNA of the total RNA was generated using the PrimeScript™ RT reagent kit (Takara Biotechnology Co., Ltd., San Jose, CA, USA). After this, RT-qPCR was performed using the SYBR-Green qPCR kit (Thermo Fisher Scientific) according to the instructions provided by the manufacturer. RT-qPCR was conducted using the following primer sequences: miR-138-5p (forward 5′-CCAGCGTGAGCTGGTGTTGTGAATC-3′ and reverse 5′-AGCAGGGTCCGAGGTATTC-3′). U6 was used as a reference gene. The RT-qPCR experiments were performed on an Applied Biosystems 7900HT Fast Real-time PCR system (Applied Biosystems; Thermo Fisher Scientific). The relative expression levels were calculated using the $2^{-\Delta\Delta Cq}$ method (*Livak & Schmittgen, 2001*) and normalized to those of the internal reference gene U6.

## Cell culture and transfection

IEC-6 cells were obtained from Beijing Beina Chuanglian Biotechnology Research Institute (Beijing, China) and were cultured at 37 °C with 5% $CO_2$. Cells were cultured in Dulbecco's modified Eagle's medium with 4 mM L-glutamine, adjusted to contain 1.5 g/L sodium bicarbonate and 4.5 g/L glucose, supplemented with 10% fetal bovine serum (Gibco, USA). To construct the HMGB1 overexpression vector (oe-HMGB1), the HMGB1 full-length sequence was inserted into pcDNA3.1 (Invitrogen, Waltham, MA, USA). NLRP3 siRNA (si-NLRP3), miR-138-5p mimic, inhibitor and corresponding negative controls were synthesized from RiboBio (Guangzhou, China). IEC-6 cells ($1 \times 10^5$) were transfected with si-NLRP3, oe-HMGB1, miR-138-5p mimic, inhibitor or corresponding negative controls using Lipofectamine® 2000 (Invitrogen; Thermo Fisher Scientific) following the manufacturer's instructions. After transfection for 48 h, the efficiency of transfection was detected by RT–qPCR and western blot analysis.

## Cell viability

IEC-6 cell viability was assessed using the Cell Counting Kit-8 (CCK-8, Beyotime, Shanghai, China). Briefly, cells were seeded in 96-well plates and treated with experimental requirements.

CCK-8 solution was added to each well. Absorbance was detected using an enzyme marker (BioTeke, Beijing, China).

## Detection of cell apoptosis

Assays for cell apoptosis were carried out using an Annexin V fluorescein isothio-cyanate/propidine iodide (Solarbio, Beijing, China). Briefly, a cold PBS buffer was used to collect IEC-6 cells, followed by 15 min of culture with Annexin VFITC/PI (1:1) at room temperature. Subsequently, flow cytometry analysis was performed with a FACS Verse flow cytometer (Becton Dickinson Biosciences, NJ, USA) and FlowJo software (version 10; Treestar, OR, USA).

## Luciferase reporter analysis

PmirGLO-NLRP3-wild (NLRP3-WT)/-mutant (NLRP3-MUT) type reporter plasmids were provided by Shanghai GenePharma Co., Ltd. HEK293 cells ($2 \times 10^5$/well) were co-transfected with NLRP3-WT/-MUT plasmid and miR-NC mimic or miR-138-5p mimic using Lipofectamine 2000 reagent (Invitrogen) at 37 °C. At 48 h post-transfection, luciferase activity was determined using the dual-luciferase reporter assay system (Promega Corporation, Madison, WI, USA). Firefly luciferase activities were normalized to Renilla luciferase activities.

## Statistical analysis

All experiments were performed in triplicate. Statistical analysis was performed using GraphPad Prism 7 (GraphPad Software Inc., San Diego, CA, USA). The results are expressed as the mean $\pm$ standard deviation (SD). Two groups were compared with Student's t test, and three or more treatments or groups were compared with one-way ANOVA followed by Tukey–Kramer post hoc analysis. A $P$ value $<0.05$ was defined as statistically significant.

## Experimental plan

The ASMVT rat model was established by artificial surgery. IEC-6 was chosen as the experimental cell line. Dual-luciferase reporter assays prove that miR-138-5p targets NLRP3. Transcriptome sequencing, RT–qPCR, western blot, immunofluorescence and tissue staining were used to confirm that miR-138-5p inhibited pyroptosis of intestinal cells, promoted intestinal epithelial tight junctions and alleviated ASMVT injury-induced intestinal barrier disruption *via* the NLRP3/HMGB1 axis.

# RESULTS

### ASMVT injury induced intestinal barrier disruption

To determine the impacts of ASMVT injury on intestinal barrier disruption, intestinal tissues were tested by histopathological examination. Figure 1A shows a pictorial of the ASMVT injury induction method. H&E staining showed notable damage to the intestinal structure, and denuded villi with exposed lamina propria and dilated capillaries were observed in the intestinal histopathology of the ASMVT group (Fig. 1B). Western blot results showed that pyroptosis-related proteins (NLRP3, ASC, cleaved caspase 1, GSDMD-N, IL-18 and IL-1$\beta$) were elevated after ASMVT injury (Fig. 1C). The tight junctions between intestinal epithelial cells are a key part of intestinal permeability, so we examined the expression of tight junction-related proteins. Western blot and immunofluorescence results showed that the levels of ZO-1, occludin and Claudin-18 were reduced after ASMVT injury (Figs. 1D and 1E). Accumulating evidence shows that HMGB1 modulates the epithelial barrier (*Kodera et al., 2020*; *Miyakawa et al., 2020*; *Peng et al., 2020*), which is increased in intestinal tissues after ASMVT injury (Fig. 1F). To identify miRNAs involved in intestinal barrier disruption, we conducted RNA-Seq analysis of intestinal tissues from ASMVT mice. Sixty miRNAs were aberrantly expressed in normal intestinal tissues compared with ASMVT-induced intestinal tissues (Figs. S1A

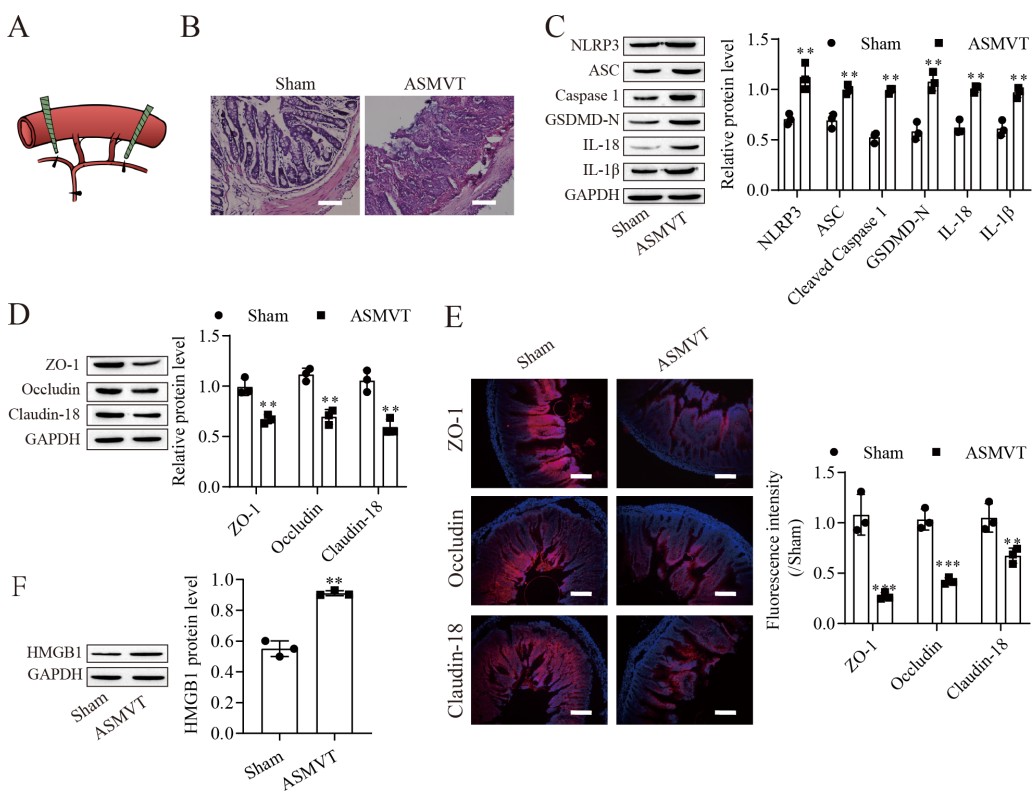

**Figure 1** **ASMVT injury induced intestinal barrier disruption.** (A) The primary branches and bilateral marginal veins of the superior mesenteric vein were ligated. (B) Intestinal tissue structure was observed by H&E staining (scale bar $=100\,\mu$m, $n = 3$). (C, D, F) The expression of pyroptosis-related proteins, tight junction-related proteins and HMGB1 was detected by western blotting ($n = 3$). (E) The expression of ZO-1, occludin and Claudin-18 was detected by immunofluorescence ($n = 3$) (scale bar $=200\,\mu$m). **$P <$ 0.01, *** $P < 0.001$, compared with the Sham group.

and S1B). The enrichment-based clustering analyses (Gene Ontology and KEGG pathway) of differentially expressed miRNA-related target genes are presented in Figs. S1C and S1D. The genes were associated with the cytoplasm, cytosol, protein binding, nucleoplasm and nucleus (Fig. S1C). The pathways of pathways in cancer, MAPK signaling pathway, mTOR signaling pathway, proteoglycans in cancer and neurotrophin signaling pathway were enriched in the differentially expressed miRNA-related target genes (Fig. S1D). Among these differentially expressed miRNAs, we focused on miR-138-5p. miR-138-5p is the only differentially expressed target of NLRP3. Therefore, we focused on the miR-138-5p/NLRP3 axis for detailed research into its role in intestinal barrier disruption in ASMVT injury.

## miR-138-5p directly targets NLRP3

The binding sites between miR-138-5p and NLRP3-WT/MUT are shown in Fig. 2A. The expression of miR-138-5p was reduced in the intestinal tissues of ASMVT injury (Fig. 2B). To confirm whether miR-138-5p could bind to the 3′UTR of NLRP3. First, we overexpressed miR-138-5p through transfection of a miR-138-5p mimic (Fig. 2C). Subsequently, we cotransfected NLRP3-WT/MUT and miR-138-5p mimic into HEK293

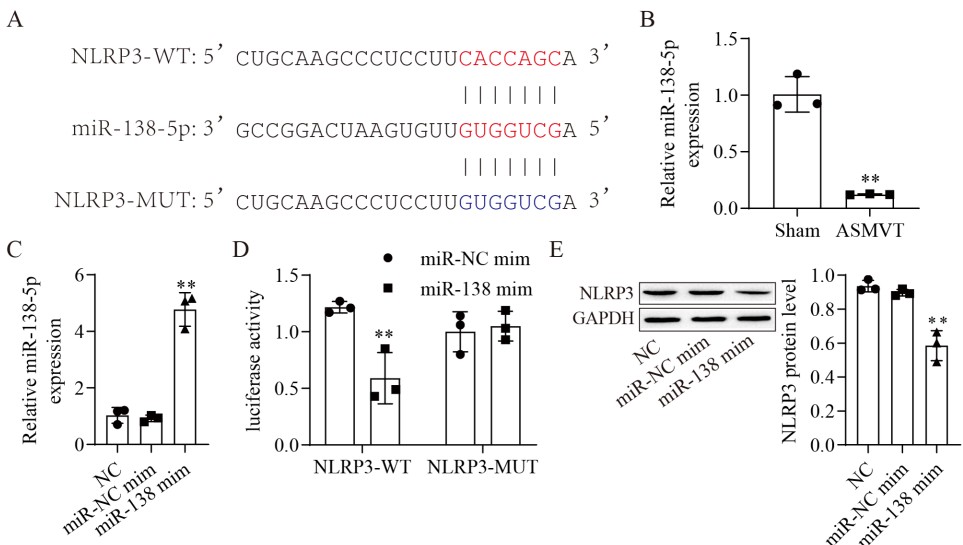

**Figure 2   MiR-138-5p directly targets NLRP3.** (A) Starbase predicted that NLRP3 targets miR-138-5p (https://starbase.sysu.edu.cn/). (B) The expression of miR-138-5p was detected by RT–qPCR in internal tissues ($n = 3$). (C) The expression of miR-138-5p was detected by RT–qPCR in HEK293 cells ($n = 3$). (D) Dual-luciferase reporter assays prove that miR-138-5p targets NLRP3 in HEK293 cells ($n = 3$). (E) NLRP3 was detected by western blot in IEC-6 cells ($n = 3$). *** $P < 0.001$, compared with Sham or miR-NC mimics.

cells and found that miR-138-5p mimic decreased the luciferase activity of NLRP3-WT in 293T cells but had no effects on NLRP3-MUT compared to the NC mimic group (Fig. 2D). Overexpression of miR-138-5p significantly reduced NLRP3 protein expression (Fig. 2E). Altogether, the aforementioned results indicated that miR-138-5p directly targeted NLRP3 genes.

## Suppression of miR-138-5p promotes NLRP3-related pyroptosis and destroys tight junctions between IEC-6 cells

We next downregulated miR-138-5p in IEC-6 cells and established miR-138-5p stably downregulating cell lines (Fig. 3A). Suppression of miR-138-5p dramatically decreased the viability (Fig. 3B) and increased the apoptosis (Fig. 3C) of IEC-6 cells. Notably, miR-138-5p knockdown dramatically increased pyroptosis-related protein levels (Fig. 3D). Western blot results showed that the level of HMGB1 was increased after miR-138-5p knockdown and that the levels of ZO-1, occludin and Claudin-18 were reduced (Fig. 3E). Immunofluorescence results showed that suppression of miR-138-5p increased NLRP3 levels and decreased tight junction-related protein levels in IEC-6 cells (Fig. 3F). Taken together, these data indicate that miR-138-5p plays a critical role in pyroptosis and tight junctions between IEC-6 cells.

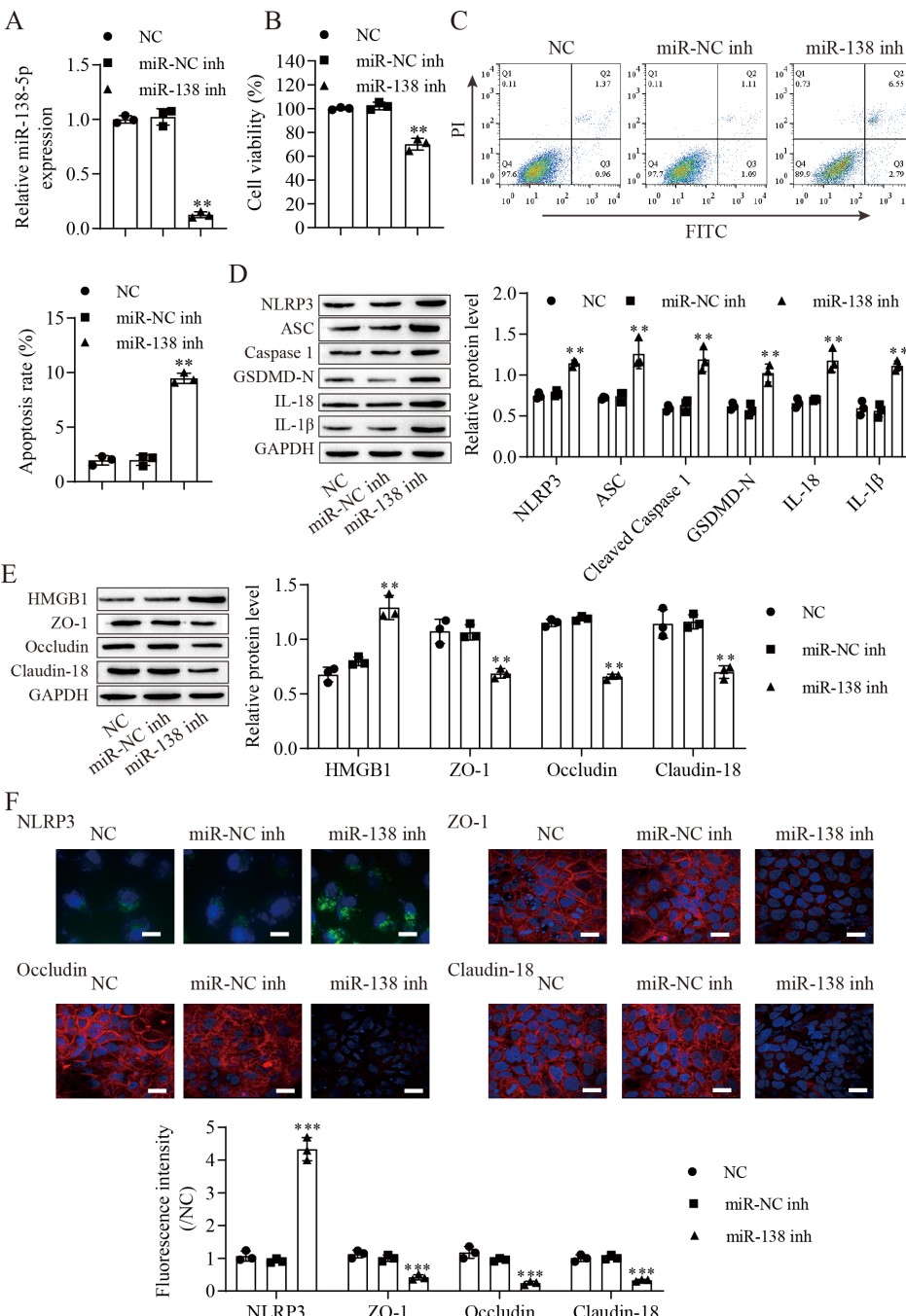

**Figure 3 Suppression of miR-138-5p promotes NLRP3-related pyroptosis and destroys tight junctions between IEC-6 cells.** (A) The expression of miR-138-5p was detected by RT–qPCR in IEC-6 cells ($n = 3$). (B) IEC-6 cell viability detected by CCK-8 ($n = 3$). (C) Cell apoptosis was measured by FITC/PI detection ($n = 3$). (D, E) Western blot analysis of the expression of pyroptosis-related proteins, tight junction-related proteins and HMGB1 ($n = 3$). (F) The expression of NLRP3, ZO-1, occludin and Claudin-18 was detected by immunofluorescence in IEC-6 cells ($n = 3$) (scale bar =20 μm). $^{**}P < 0.01$, $^{***}P < 0.001$, compared with miR-NC inh.

## miR-138-5p targets NLRP3 expression to regulate NLRP3-related pyroptosis and tight junctions in IEC-6 cells

To explore the effect of the miR-138-5p/NLRP3 axis on NLRP3-related pyroptosis and tight junctions between IEC-6 cells. IEC-6 cells were transfected with si-NLRP3 to downregulate NLRP3 expression. si-NLRP3-2 achieved more effective knockdown efficiency (Fig. 4A). The results indicated that suppression of miR-138-5p inhibited cell viability but was reversed with knockdown of NLRP3 (Fig. 4B). In contrast, cell apoptosis was elevated after miR-138-5p downregulation, which was reversed with NLRP3 knockdown (Fig. 4C). The levels of pyroptosis-related proteins were significantly increased by downregulating miR-138-5p but decreased with knockdown of NLRP3 (Fig. 4D). HMGB1 protein levels were increased by downregulating miR-138-5p, which was decreased with knockdown of NLRP3 (Fig. 4E). In contrast, the levels of ZO-1, occludin and Claudin-18 were reduced by downregulating miR-138-5p but elevated after knockdown of NLRP3. Immunofluorescence results showed that suppression of miR-138-5p increased NLRP3 levels and decreased tight junction-related protein levels in IEC-6 cells, and these effects were reversed by NLRP3 knockdown (Fig. 4F). Therefore, these results indicated that miR-138-5p regulates NLRP3-related pyroptosis and tight junctions between IEC-6 cells by targeting NLRP3 expression.

## miR-138-5p regulates tight junctions between IEC-6 cells *via* the NLRP3/HMGB1 axis

To investigate the effect of HMGB1 on the miR-138-5p/NLRP3 axis, tight junctions between IEC-6 cells were regulated. We next overexpressed HMGB1 in IEC-6 cells and established HMGB1 stably overexpressing cell lines (Fig. 5A). Western blot results showed that the levels of ZO-1, occludin and Claudin-18 were reduced after miR-138-5p knockdown, which was reversed with knockdown of NLRP3, but they were finally repressed by upregulating HMGB1 (Fig. 5B). Similar results were also observed in IEC-6 cells by immunofluorescence staining (Fig. 5C). Therefore, these results indicated that miR-138-5p regulates tight junctions between IEC-6 cells *via* the NLRP3/HMGB1 axis.

## miR-138-5p ameliorates ASMVT injury-induced intestinal barrier disruption

To further explore the role of miR-138-5p in intestinal barrier disruption after ASMVT injury, we overexpressed miR-138-5p in ASMVT mice (Fig. 6A). H&E staining showed that overexpression of miR-138-5p ameliorated ASMVT injury-induced intestinal structure damage (Fig. 6B). Western blot results showed that the levels of pyroptosis-related proteins were significantly increased but were inhibited with overexpression of miR-138-5p (Fig. 6C). The level of HMGB1 was increased after ASMVT injury but was reversed with overexpression of miR-138-5p (Fig. 6D). In contrast, the levels of ZO-1, occludin and Claudin-18 were reduced after ASMVT injury and were increased with miR-138-5p overexpression. Immunofluorescence results showed that the expression of NLRP3 and HMGB1 was increased after ASMVT injury but was reversed with overexpression of miR-138-5p (Fig. 6E). In contrast, the expression of ZO-1, occludin and Claudin-18 was reduced after ASMVT injury and was increased with miR-138-5p overexpression.

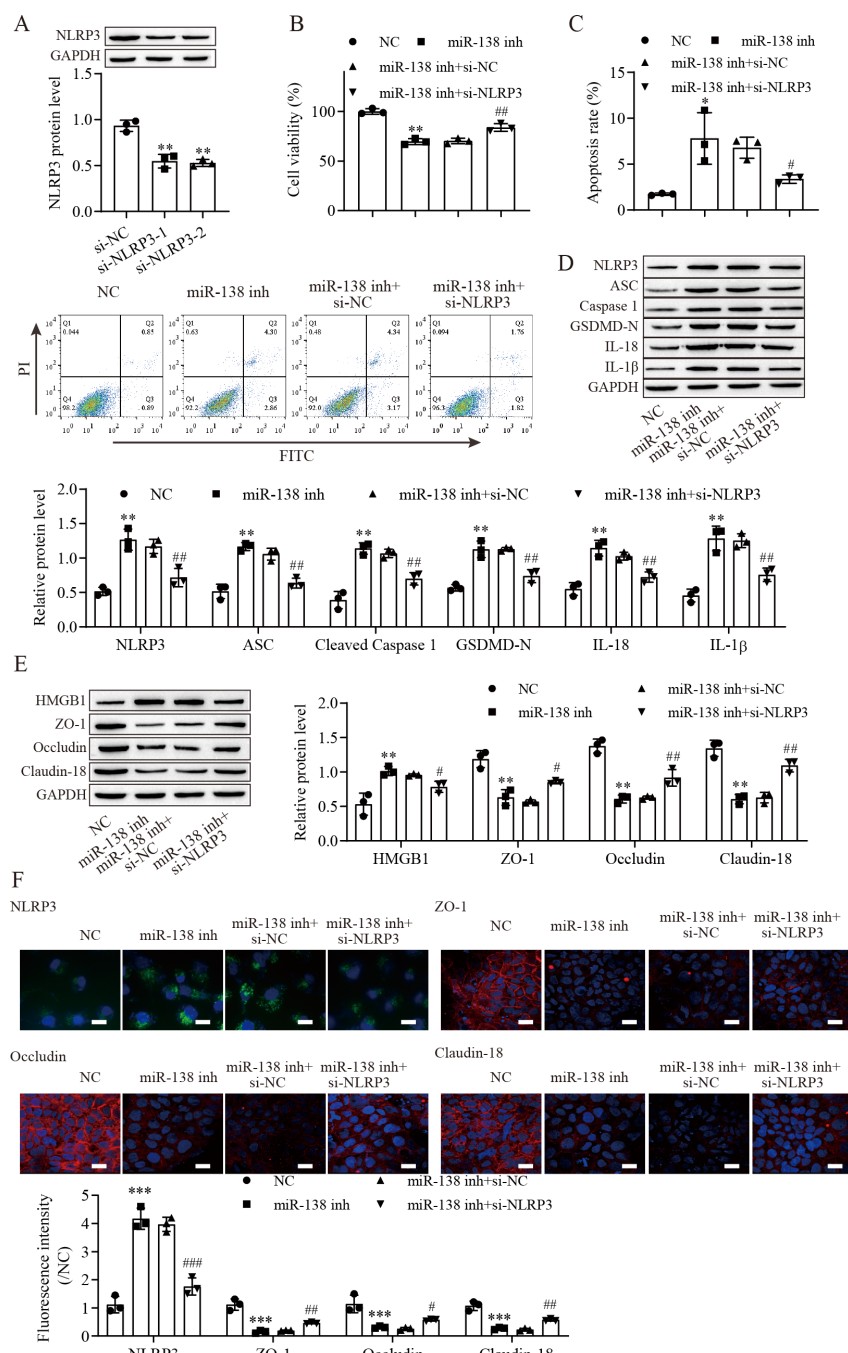

**Figure 4 MiR-138-5p targets NLRP3 expression to regulate NLRP3-related pyroptosis and tight junctions in IEC-6 cells.** (A, D, E) The expression of pyroptosis-related proteins, tight junction-related proteins and HMGB1 was detected by western blot ($n = 3$). (B) IEC-6 cell viability detected by CCK-8 ($n = 3$). (C) Cell apoptosis was measured by FITC/PI detection ($n = 3$). (F) The expression of NLRP3, ZO-1, occludin and Claudin-18 was detected by immunofluorescence in IEC-6 cells ($n = 3$) (scale bar $=20\,\mu$m). ** $P < 0.01$, *** $P < 0.001$, compared with NC; # $P < 0.05$, ## $P < 0.01$, ###$P < 0.001$, compared with miR-138 inhi+si-NC.

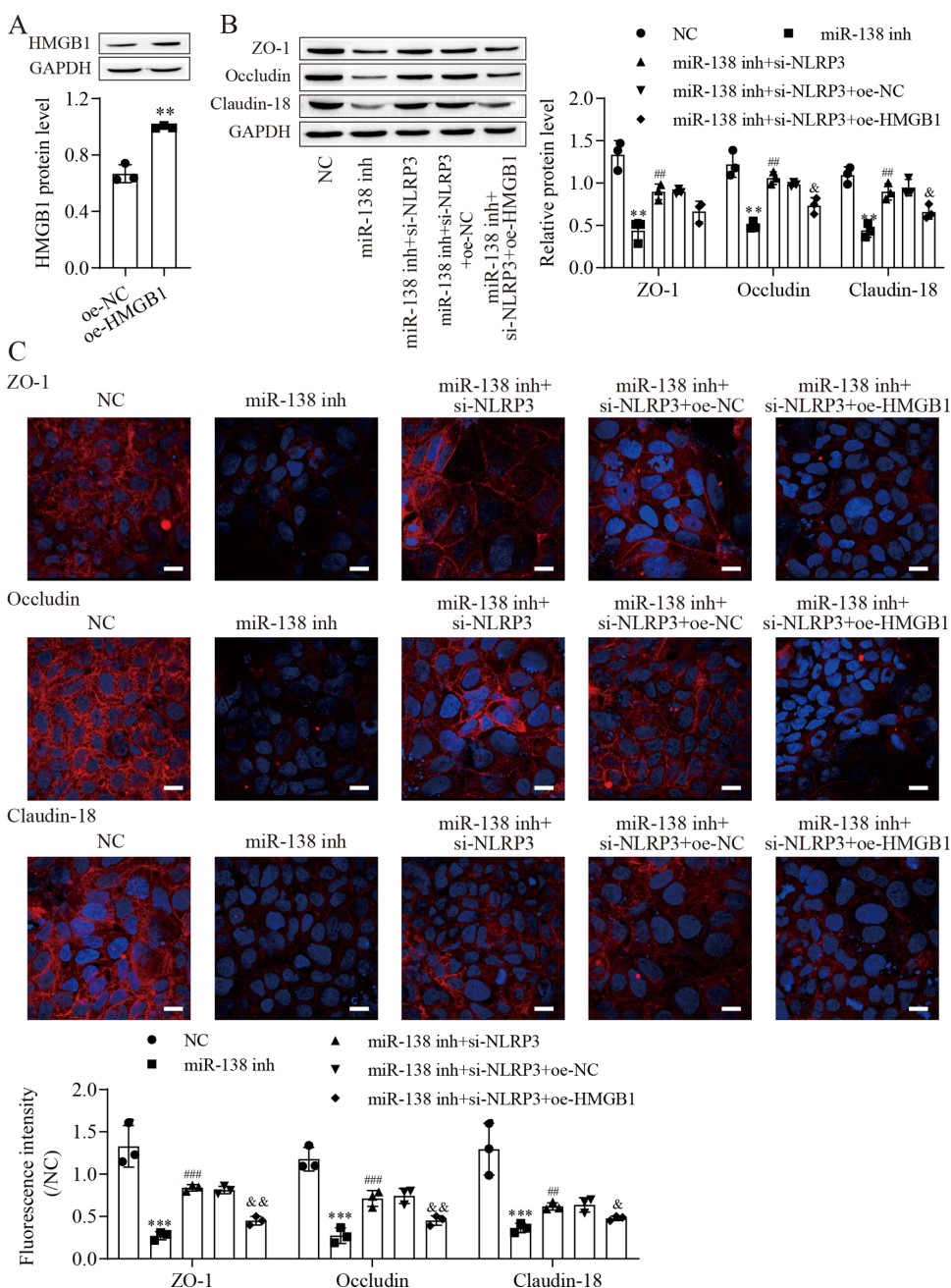

**Figure 5  MiR-138-5p regulates tight junctions between IEC-6 cells *via* the NLRP3/HMGB1 axis.** (A, B) The expression of HMGB1 and tight junction-related proteins was detected by western blot ($n = 3$). (C) The expression of ZO-1, occludin and Claudin-18 was detected by immunofluorescence in IEC-6 cells ($n = 3$) (scale bar $=20$ μm). $^{**}P < 0.01$, $^{***}P < 0.001$, compared with oe-NC or NC; $^{\#\#}P < 0.01$, $^{\#\#\#}P < 0.001$, compared with miR-138 inh; $^{\&}P < 0.05$, $^{\&\&}P < 0.01$, compared with miR-138 inhi+si-NLRP3+oe-NC.

Therefore, these results indicated that miR-138-5p ameliorates ASMVT injury-induced intestinal barrier disruption by inhibiting NLRP3-related pyroptosis and promoting tight junctions between intestinal epithelial cells.

ASMVT: Acute superior mesenteric venous thrombosis; H&E: hematoxylin and eosin; miRNAs: microRNAs; miR-138: miR-138-5p; mim: mimic; inh: inhibitor.

## DISCUSSION

miRNAs play important roles in the initiation and progression of thrombotic diseases (*Bijak et al., 2016*; *Tay et al., 2018*; *Vijay et al., 2019*). We observed that few studies determined the abnormal expression of miRNAs during ASMVT. In our study, we first determined that ASMVT injury leads to changes in miRNA expression. Through RNA-Seq analysis, we identified 60 differentially expressed miRNAs after ASMVT. Here, we found that a significant number of these annotated proteins were enriched in the cellular component of cell junctions, which is important in the intestinal epithelial barrier (*Luissint, Parkos & Nusrat, 2016*). KEGG pathway analysis revealed that a significant number of these annotated proteins were enriched in the mTOR and MAPK signaling pathways, which have important roles in pyroptosis (*Li et al., 2018*; *Li et al., 2019*) and the intestinal epithelial barrier (*Shao et al., 2017*; *Xiong et al., 2020*), and the chemokine signaling pathway, which affects the intestinal epithelial barrier. GO terms and KEGG pathway analysis suggested that the differential expression of miRNAs is closely related to the intestinal epithelial barrier. Among these lowly expressed miRNAs in ASMVT, we defined a miRNA, miR-138-5p, that plays a critical role in intestinal epithelial barrier disruption of ASMVT, and overexpression of miR-138-5p relieves ASMVT injury-induced intestinal epithelial barrier disruption.

Pyroptosis is a form of programmed cell death that is associated with inflammatory product release. Intestinal ischemia −reperfusion (I/R) injury induces intestinal inflammation and activation of NLRP3-related pyroptosis (*Jia et al., 2020*). In this study, we found that pyroptosis-related proteins (ASC, cleaved caspase 1 and GSDMD-N) and NPRP3, which are considered important activators of pyroptosis, were upregulated, indicating the presence of pyroptosis (*He et al., 2015*; *Shi et al., 2015*; *Sun et al., 2019*). Notably, we also found that the levels of IL-18 and IL-1$\beta$, which are commonly detected during pyroptosis, were elevated (*Swanson, Deng & Ting, 2019*). Intestinal I/R injury leads to disruption of the intestinal barrier and intestinal cell death by inducing intestinal inflammation and activation of NLRP3-related pyroptosis (*Jia et al., 2020*). We found denuded villi with exposed lamina propria and dilated capillaries in the intestinal histopathology of the ASMVT group. Interestingly, HMGB1 protein was significantly increased in the intestine after ASMVT injury. HMGB1 can mediate the proinflammatory responses of cells and induce apoptotic, autophagic or pyroptotic cell death (*Geng et al., 2015*; *Xu et al., 2014*). Our data support the idea that pyroptosis-mediated intestinal cell death and ASMVT injury play a critical role in the disruption of the intestinal barrier. We observed that few studies have revealed the role of pyroptosis during ASMVT injury.

Accumulating evidence shows that the NLRP3 inflammasome is considered an important activator of pyroptosis (*Hruda & Julsrud, 1989*). The NLRP3 protein recruits the adapter

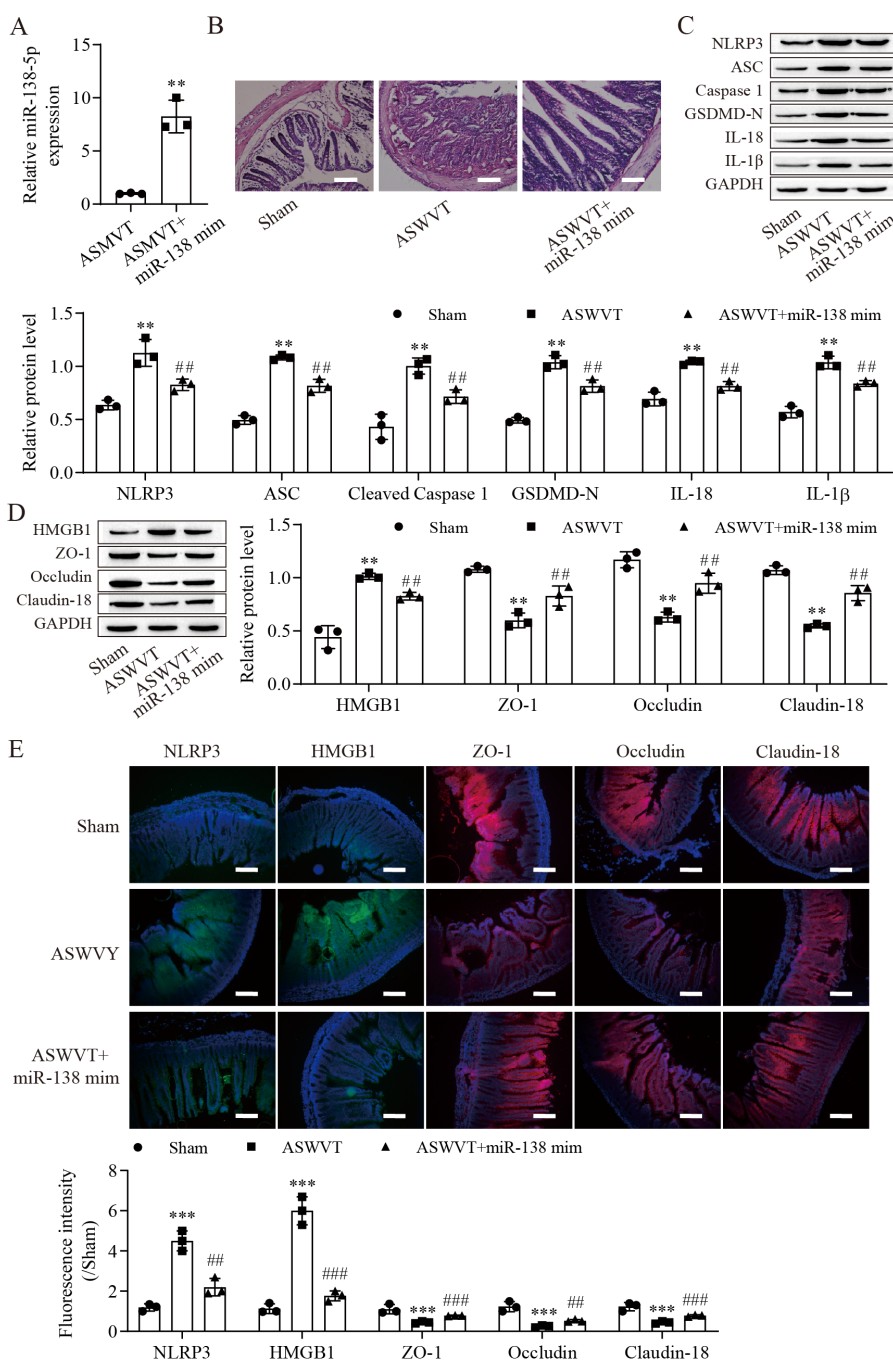

**Figure 6** **MiR-138-5p ameliorates ASMVT injury-induced intestinal barrier disruption.** (A) The expression of miR-138-5p was detected by RT–qPCR in intestinal tissues ($n = 3$). (B) The intestinal tissue structure was observed by H&E staining ($n = 3$) (scale bar $=100\,\mu$m). (C, D) The expression of pyroptosis-related proteins, tight junction-related proteins and HMGB1 was measured by western blot in intestinal tissues ($n = 3$). (E) The expression of NLRP3, ZO-1, occludin and Claudin-18 was detected by immunofluorescence in intestinal tissues ($n = 3$) (scale bar $=200\,\mu$m). $^{**}P < 0.01$, $^{***}P < 0.001$, compared with Sham; $^{\#\#}P < 0.01$, $^{\#\#\#}P < 0.001$, compared with ASMVT. ASMVT: Acute superior mesenteric venous thrombosis; H&E, hematoxylin and eosin; miRNAs, microRNAs; miR-138, miR-138-5p; mim, mimic; inh, inhibitor.

ASC protein, which recruits procaspase-1, resulting in its cleavage and activation, and then triggers cleavage of GSDM family proteins (*He et al., 2015*; *Zahid et al., 2019*). We found that NLRP3 protein was increased in the intestine after ASMVT injury. Similar to our results, *Jia et al. (2020)* reported that intestinal I/R injury significantly elevated NLRP3 protein levels. The competing endogenous RNA hypothesis is an important mechanism that allows lncRNAs/circRNAs to sponge miRNAs to modulate mRNAs (*Zhong et al., 2018*). Here, we found that miR-138-5p regulated NLRP3 expression by binding to the 3′ UTR of NLRP3 mRNA molecules, leading to NLRP3 degradation. Accumulating evidence shows that NLRP3 is a direct target of miR-138-5p in multiple cells (*Feng et al., 2021a*; *Feng et al., 2021b*; *Luo et al., 2021*). In our study, significant pyroptosis and apoptosis in IEC-6 cells were promoted, and viability was decreased by downregulating miR-138-5p levels, and the effects of miR-138-5p downregulation were reversed by knocking down NLRP3 levels. Moreover, we also found that overexpression of miR-138-5p inhibited NLRP3-related pyroptosis in the intestine after ASMVT injury. Previous studies have reported that miR-138-5p regulates pyroptosis by targeting sirtuin 1, NLRP3 and sestrin 2 (*An et al., 2021*; *Luo et al., 2021*; *Mao et al., 2019*). This study indicates that miR-138-5p is critical for sustaining NLRP3-related pyroptosis during ASMVT injury.

NLRP3-mediated cell pyrolysis promotes inflammatory factor release, causes intestinal injury, and downregulates the expression of cell tight junction-related proteins, such as ZO-1, occludin and Claudin-18, resulting in intestinal mucosal damage (*Lin et al., 2021*). Here, we show that cell tight junction-related proteins were decreased in the intestine after ASMVT injury. Accumulating evidence shows that HMGB1 modulates the epithelial barrier, including the airway, intestine and cervix (*Kodera et al., 2020*; *Miyakawa et al., 2020*; *Peng et al., 2020*). NLRP3 inflammasome activation triggers the release of the permeability factor HMGB1, thereby destroying tight junctions and increasing endothelial permeability (*Chen et al., 2015a*; *Chen et al., 2016*). In this study, knockdown of NLRP3 decreased HMGB1 levels in IEC-6 cells and increased cell tight junction-related protein levels, and the levels of cell tight junction-related proteins were reversed by overexpression of HMGB1. Moreover, overexpression of miR-138-5p inhibited HMGB1 expression and increased cell tight junction-related protein expression in the intestine after ASMVT injury. Similar to our results, *Chen et al. (2015b)* reported that visfatin-induced cell tight junction-related protein downregulation in endothelial cells was attributed to high HMGB1 release, which was abolished by silencing the NLRP3 gene. These results indicated that the miR-138-5p-mediated NLRP3/HMGB1 axis in cell tight junctions plays a critical role in the regulation of the intestinal epithelial barrier.

## CONCLUSIONS

miR-138-5p expression was downregulated in ASMVT-induced intestinal tissues. Overexpression of miR-138-5p inhibited HMGB1 expression by targeting NLRP3, inhibited pyroptosis, promoted intestinal epithelial intercellular tight junctions and ameliorated ASMVT injury-induced intestinal barrier disruption. This provides a new idea and method

for the study of intestinal barrier disruption to improve potential biomarkers and to treat it.

### Funding

This work was supported by the Chen Zhong Expert Workstation Fund of Yunnan Province (2005AF150018), the National Natural Science Foundation of China (81960447), the Open project of National Health Commission key laboratory of drug addiction medicine (2020DAMOP-0014), the Yunnan Medical Reserve Talents Project (H-2018063), the Opening project of Clinical Medical Center of The First People's Hospital of Yunnan Province (2021LCZXXF-XG02) and the Doctoral Research Fund of The First People's Hospital of Yunnan Province (20206023). The funders had no role in study design, data collection and analysis, decision to publish, or preparation of the manuscript.

### Grant Disclosures

The following grant information was disclosed by the authors:
Chen Zhong Expert Workstation Fund of Yunnan Province: 2005AF150018.
National Natural Science Foundation of China: 2020DAMOP-0014.
Yunnan Medical Reserve Talents Project: H-2018063.
The Opening project of Clinical Medical Center of The First People's Hospital of Yunnan Province: 2021LCZXXF-XG02.
Doctoral Research Fund of The First People's Hospital of Yunnan Province: 20206023.

### Competing Interests

The authors declare there are no competing interests.

### Author Contributions

- Yuejin Li conceived and designed the experiments, authored or reviewed drafts of the article, and approved the final draft.
- Ping Ling conceived and designed the experiments, authored or reviewed drafts of the article, and approved the final draft.
- Yu Li performed the experiments, prepared figures and/or tables, and approved the final draft.
- Yongzhi Wang performed the experiments, prepared figures and/or tables, and approved the final draft.
- Guosan Li performed the experiments, prepared figures and/or tables, and approved the final draft.
- Changtao Qiu analyzed the data, prepared figures and/or tables, and approved the final draft.
- Jianghui Wang analyzed the data, prepared figures and/or tables, and approved the final draft.
- Kunmei Gong conceived and designed the experiments, authored or reviewed drafts of the article, and approved the final draft.

## Animal Ethics

The following information was supplied relating to ethical approvals (*i.e.*, approving body and any reference numbers):

The animal experiments have been approved by the Animal Ethics Committee of Kunming University of Science and Technology

## DNA Deposition

The following information was supplied regarding the deposition of DNA sequences:

The sequencing data is available at GEO: GSE240523.

## Data Availability

The raw data are available in the Supplemental Files.

## Supplemental Information

Supplemental information for this article can be found online at http://dx.doi.org/10.7717/peerj.16692#supplemental-information.

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
