# Peer review of "miR-138-5p ameliorates intestinal barrier disruption caused by acute superior mesenteric vein thrombosis injury by inhibiting the NLRP3/HMGB1 axis"

_PeerJ, doi:10.7717/peerj.16692_

## Round 0.1 · original submission · Major Revisions

The figures (figure 1A, 6B etc) as mentioned by reviewer 2 should be replaced with other representative figures. Or mention in the writing.

Please add better representative images for the Western Blot better which will better match with the quantification and please show the data points in the quantification graphs.

**Language Note:** The review process has identified that the English language must be improved. PeerJ can provide language editing services - please contact us at [email protected] for pricing (be sure to provide your manuscript number and title). Alternatively, you should make your own arrangements to improve the language quality and provide details in your response letter. – PeerJ Staff

·

Basic reporting

The manuscript is well-written and organized and has a significant increment to the subject.
However, I have below comments to improve the manuscripts.
The experimental plan should be discussed briefly before beginning any new results, otherwise, it would be tough to understand whatever is shown. The method must be clearly mentioned in figure legends throughout the MS.

Preparing Figure 1A showing a pictorial/cartoon of the ASMVT injury induction method would be helpful to the readers.

Experimental design

The experimental plan should be outlined briefly before discussing the results.

Validity of the findings

Individual data points should be shown in all bar graphs shown.

Reviewer 2 ·

Basic reporting

In this study the authors show that miR-138-5p was downregulated in ASMVT-induced intestinal tissues. The inhibition of miR-138-5p promoted NLRP3-related pyroptosis and disrupted the tight junctions exacerbating the ASMVT injury. Knockdown of NLRP3 reversed the inhibitory effects as well as the decrease in tight junction proteins caused by the suppression of miR-138-5p. These effects were subsequently inhibited by the overexpression of HMGB1.

Experimental design

The experiments were meticulously planned and effectively presented, and the findings from this study hold valuable relevance for both the research and clinical communities.

Validity of the findings

Although the results were well presented, there are some concerns that need to be addressed. For example, some control images have been reused at multiple instances (Eg. Fig 1B and 6B, Fig 1D and 6E, Fig.4F and 5C). Is it made clear in the text? Confirm if it is permitted by the journal.

Additional comments

Other concerns include:
1. The manuscript MUST be revised for English language (grammar, syntax and spelling) by a professional or a fluent English speaker. There are innumerable instances with wrong spellings, wrong grammar, wrong sentence construction etc. The whole manuscript requires thorough editing.
2. Provide a higher magnification image for Tight junction proteins in the tissue to visualize the localization.
3. Quantification of miRNA requires using a reference gene such as U6 or RNU, Was any reference gene used in this study (Eg for Fig. 2C)
4. Authors need to discuss/explain how NLRP3/HMGB1 axis regulate tight junction proteins.
5. In Fig 2 with transfection of miR-138-5p mimic which cells were used?
6. Explain why we see a dramatic loss of Occludin, ZO1 and Caludin 1 in immunofluorescence staining but not with WB?
7. What is AWVT in Fig. 6?
8. Is it explained how miR-138-5p was overexpressed in ASMVT mouse (Fig. 6)?

---

## Round 0.2 · accepted · Accept

Please check carefully all the typing errors and grammatical mistakes.

·

Basic reporting

All my concerns were appropriately addressed.

Experimental design

Looks good for the publication.

Validity of the findings

Results may be replicated if their method is appropriately followed.

Additional comments

The article may be accepted for the publication.

Reviewer 2 ·

Basic reporting

The authors have addressed my concerns.

Experimental design

NA

Validity of the findings

NA

Additional comments

NA